# Synergistic phenotypic adaptations of motile purple sulphur bacteria *Chromatium okenii* during lake-to-laboratory domestication

Francesco Di Nezio[1,2], Irvine Lian Hao Ong[3], René Riedel[3], Arkajyoti Goshal[3], Jayabrata Dhar[4], Samuele Roman[1,5], Nicola Storelli[1,2], Anupam Sengupta[3,6]*

**1** Department of Environment, Institute of Microbiology, Constructions and Design, University of Applied Sciences and Arts of Southern Switzerland (SUPSI), Mendrisio, Switzerland, **2** Microbiology Unit, Department of Plant Sciences, University of Geneva, Geneva, Switzerland, **3** Physics of Living Matter Group, Department of Physics and Materials Science, University of Luxembourg, Luxembourg City, Luxembourg, **4** Department of Mechanical Engineering, National Institute of Technology, Durgapur, India, **5** Alpine Biology Center Foundation, Bellinzona, Switzerland, **6** Institute for Advanced Studies, University of Luxembourg, Esch-sur-Alzette, Luxembourg

* anupam.sengupta@uni.lu

**Data Availability Statement:** All relevant data are within the paper and its Supporting Information files.

## Abstract

Isolating microorganisms from natural environments for cultivation under optimized laboratory settings has markedly improved our understanding of microbial ecology. Artificial growth conditions often diverge from those in natural ecosystems, forcing wild isolates into distinct selective pressures, resulting in diverse eco-physiological adaptations mediated by modification of key phenotypic traits. For motile microorganisms we still lack a biophysical understanding of the relevant traits emerging during domestication and their mechanistic interplay driving short-to-long-term microbial adaptation under laboratory conditions. Using microfluidics, atomic force microscopy, quantitative imaging, and mathematical modeling, we study phenotypic adaptation of *Chromatium okenii*, a motile phototrophic purple sulfur bacterium from meromictic Lake Cadagno, grown under laboratory conditions over multiple generations. Our results indicate that naturally planktonic *C. okenii* leverage shifts in cell-surface adhesive interactions, synergistically with changes in cell morphology, mass density, and distribution of intracellular sulfur globules, to suppress their swimming traits, ultimately switching to a sessile lifeform. A computational model of cell mechanics confirms the role of such phenotypic shifts in suppressing the planktonic lifeform. By investigating key phenotypic traits across different physiological stages of lab-grown *C. okenii*, we uncover a progressive loss of motility during the early stages of domestication, followed by concomitant deflagellation and enhanced surface attachment, ultimately driving the transition of motile sulfur bacteria to a sessile state. Our results establish a mechanistic link between suppression of motility and surface attachment via phenotypic changes, underscoring the emergence of adaptive fitness under laboratory conditions at the expense of traits tailored for natural environments.

**Funding:** This work was supported by the Swiss National Science Foundation (grant number 315230–179264) and by the Institute of Microbiology (IM) of the University of Applied Sciences and Arts of Southern Switzerland (SUPSI) through the financing from the Department of "Socialità e Sanità" (DSS) of the Canton Ticino. I.L. H.O. thanks the support from Marie Skłodowska-Curie Actions Individual Fellowship (BIOMIMIC grant agreement number 897629). Support of the Luxembourg National Research Fund's AFR-Grant (Grant no. 13563560), the ATTRACT Investigator Grant, A17/MS/11572821/MBRACE (to A.S.), and the FNR-CORE Grant (No. C19/MS/13719464/TOPOFLUME/Sengupta) are gratefully acknowledged.

**Competing interests:** The authors have declared that no competing interests exist.

## Introduction

The ability to isolate microorganisms from natural settings and grow them under controlled laboratory conditions has long been a critical issue [1, 2]. Nevertheless, this step was essential in advancing our current understanding of the general behavior, physiology, and fitness of microbes in a systematic manner. Natural microbial habitats offer conditions which are far from steady, wherein diverse abiotic and biotic pressures shape microbial lifestyles and survival strategies [3, 4]. Under the tailored growth conditions of laboratory environments, freshly isolated microorganisms experience a resource-replete setting, which often result in loss of key ecophysiological traits, following phenotypic adaptation or long-term mutations under favorable conditions [5–7], ultimately leading to adaptation of strains to laboratory conditions [3]. Adaption to artificial settings is characterized by changes in morphotype, physiology, and biological fitness, with the first signs of such diversification appearing within 2–3 days of domestication [5]. While most studies to date have focused on sessile species, phenotypic alterations including loss of motility and associated rapid growth, which can be attributed to the higher costs of flagellar construction, have also been observed in motile species growing in batch cultures [8–10]. Studies so far indicate that species may alter or lose multiple traits concomitantly [4, 8, 10], however little is known if the loss of a trait proceeds synergistically, or independent of other emerging traits.

Here we focus on the motile purple sulfur bacterium (PSB) *Chromatium okenii*, a member of the *Chromatiaceae* family isolated from the meromictic Lake Cadagno [11], known to produce the phenomenon of bioconvection [12]. PSB normally develop under anoxic conditions in the presence of light where sulfide ($S^{2-}$) serves as an electron donor in the photosynthetic process and is oxidized to sulfate ($SO_4^{2-}$) through an intermediate accumulation of elemental sulfur ($S^0$) within the cell in the form of sulfur globules (SGBs). SGBs, a key feature of PSB, serve as intracellular storage for sulfur and their number and size vary according to the availability of environmental reduced sulfur [13, 14]. The euxinic conditions (when water is both anoxic and sulfidic) of Lake Cadagno, a meromictic lake located in the southern Swiss Alps, provide a conducive habitat for a thriving community of anoxygenic phototrophic sulfur bacteria [15]. Within the distinctive bacterial layer (BL) of this lake, are seven PSB species, including *C. okenii*, and two green sulfur bacteria (GSB) species playing pivotal roles in the lake's major biogeochemical processes [16–18].

PSB *C. okenii* is a positively phototactic and negatively aerotactic species [19], moving towards light sources that penetrate the chemocline. This behavior is essential for optimizing photosynthetic activity, which depends on the availability of light and sulfide, and its flagellar motility provides it with a distinct advantage, through the process of bioconvection, the convection flow of a fluid induced by the density gradient (here intended as difference in weight between adjacent layers of water due to the local accumulation of microbial cells), resulting from the coordinated directional swimming of microorganisms [12]. This phenomenon allows *C. okenii* to reach the most favorable environmental niches and compete effectively with other phototrophic sulfur bacteria [20]. Despite being crucial for navigating the vertical gradients of light and sulfide in the natural habitat, maintaining a flagellar motility system is energetically costly and its regulatory efficiency is under considerable selective pressure in nature [21, 22]. If organisms can reduce their exposure to such stressors or are phenotypically prepared for anticipated changes before they occur, they may perform better and be more likely to persist [23], e.g., by transitioning to a non-motile state, increasing both cohesion among cells and adhesion to solid surfaces [24].

Here, we use a combination of microfluidics, quantitative imaging, mathematical modeling, and atomic force microscopy (AFM) to study biophysical phenotypic changes in natural

isolates of *C. okenii* compared to their domesticated counterparts, grown over multiple generations under laboratory conditions.

## Materials and methods

### *In situ* cell sampling

Sampling season in Lake Cadagno started in June (after ice melt) and ended in October 2022. The *Chromatium okenii* cells used in the present study were collected on 13 July from a platform anchored above the deepest point of the lake (21 m). Water for biological analysis was sampled from the chemocline through a Tygon tube (20 m long, inner diameter 6.5 mm, volume 0.66 l) at a flow rate of $1l \ min^{-1}$ using a peristaltic pump (KNF Flodos AG, Sursee, Switzerland). Samples were kept refrigerated as to maintain the temperature at which they were sampled (4°C) and in the dark and analyzed for microbiological parameters within 1 h after sampling.

The Alpine Biology Center, Piora (CBA) approved the field site access and granted permission for the fieldwork carried out.

### Laboratory cell culture

Purple sulfur bacterium *Chromatium okenii* strain LaCa, isolated from the chemocline of Lake Cadagno and grown in the laboratory since 2016 [11], served as domesticated strain for the study. Cells are cultivated in Pfennig's medium I [25] prepared in a 2.0 l bottle using a flushing gas composition of 90% $N_2$ and 10% $CO_2$ according to Widdel and Bak [26], reduced by adding a neutralized solution of $Na_2S \ x \ 9H_2O$ to a concentration of 1.0 mM $S^{2-}$ and then adjusted to a pH of approximately 7.1. Cells are cultured in 100 ml sterile serum bottles at 20°C temperature in a diurnal growth chamber (SRI21D-2, Sheldon Manufacturing Inc., Cornelius, OR, USA) under a light/dark photoperiod of 16/8 h and a light intensity of 38.9 $\mu mol \ m^{-2} \ s^{-1}$ PPFD (Photosynthetic Photon Flux Density), within the photosynthetic active radiation range (400–700 nm). Cultures used for swimming properties and phenotypic traits quantification experiments were propagated from a 35/40-day old pre-culture (stationary growth stage) to standardize the starting population physiological status. The experiments were carried out within a fixed period of the day (between 08:30 h and 13:00 h) to rule out any potential artefacts due to possible circadian cycles of *C. okenii*. The specific growth rate was calculated as the rate of increase in the cell population per unit of time (hours). To investigate the effect of adaptation to artificial settings, we used *C. okenii* cells sampled from the lake and we compared them with cells cultivated in the artificial setting of the laboratory incubator under artificial light (domesticated) (S1A Fig in S1 File). Laboratory-grown *C. okenii* cultures were analyzed in their exponential growth stage so to have metabolically active cells providing a better insight into how domesticated and wild cells show different adaptation to their respective environments. Experiments were conducted in triplicates, with replicates taken from different vials. The main cell features used to describe *C. okenii* morphology are shown in S1B Fig in S1 File.

### Flow cytometry

*C. okenii* natural and domesticated cells strain LaCa were monitored by flow cytometry (FCM) measuring chlorophyll-like autofluorescence particle events. Cell counting was performed on a BD Accuri C6 Plus cytometer (Becton Dickinson, San José, CA, USA), as described in Danza *et al.* [27]. The cytometer FL3 interference filter detects emissions above 670 nm. This allowed us to monitor fluorescence from both Bacteriochlorophyll *a* and *b* (emission peak ~ 750–800 nm) and from okenone (~ 650–750 nm). In conjunction with the fluorescence signature, PSB

*C. okenii* can be distinguished from the other anoxygenic phototrophic sulfur bacteria inhabiting the BL of Lake Cadagno based on morphological characteristics [27].

## Cell tracking

To quantify *C. okenii* cell motility, movies were recorded at 10 frames per second for 10 s and converted to image sequences. Details of cell tracking are reported in S1 Text in S1 File.

## Volume quantification of intracellular SGBs

To characterize and quantify the biosynthesis and accumulation of sulfur globules (SGBs), cells were sampled from the culture bottles at different time intervals to cover the whole exponential growth stage. To identify and characterize the accumulation of SGBs in single cells, phase contrast and fluorescence microscopy with high-resolution color camera imaging, was carried out. Images were acquired using a Hamamatsu ORCA-Flash camera (1 μm = 10.55 pixels) coupled to an inverted microscope (Olympus CellSense LS-Ixplore) with a 100x oil objective. Overall, this gave a resolution of 0.06 μm, allowing us to precisely identify and characterize the SGBs accumulating within single cells (S2A Fig in S1 File). To extract *C. okenii* cell area and SGBs number and dimension (size and volume), pictures and movies of single cells were acquired and analyzed as described in Sengupta *et al.* [28, 29].

## Cell morphology and flagellar position

Phase contrast (Zeiss AxioScope A1 epifluorescence microscope) and scanning electron microscopy (Phenom XL G2 Desktop SEM, Thermo Scientific, Waltham, MA, USA) were used to quantify cell morphological characteristics and determine the position of the flagella of *C. okenii*. For SEM imaging, samples were prepared as described in Relucenti *et al.* [30]. Cells were sampled from the upper part of the culture vials, to have them as actively motile as possible and exclude non-motile ones, which sedimented at the bottom. Morphological features, such as aspect ratio and volume, were derived from the contour area extracted by thresholding and ImageJ image analysis. Overall, flagellated cells indicate that they execute pusher type swimming [31].

## Quantification of cellular mass density

To quantify the influence of SGBs on cell density we assumed that the density of structural cell material and the density of the sulfur globules remained constant over the course of the experiment. Other inclusions (i.e., PHB, glycogen) were either undetected or present at a constant quantity and thus considered as components of the cell's structural material (here cytoplasm) (S2 Text in S1 File).

## Cell phototactic behavior

To investigate the response of *C. okenii* to light, swimming cells sampled from the lake were loaded into rectangular millimetric chambers (microfluidic ChipShop GmBH, Jena, Germany), incubated in the dark for 1 h and then exposed to diffused, low-intensity light from a cold white LED array source (Thorlabs GmBH, Bergkirchen, Germany) placed above the chamber. The cold white spectrum shows peaks at approximately 450–470 nm (blue) and 520–570 nm (green). These wavelengths reach the BL depth in Lake Cadagno and are harvested by *C. okenii* for photosynthesis [32]. As a first step, after a 60 min incubation in the dark, one half of the chamber was covered with aluminum foil and the other half was left exposed to light at 14 cm from the LED source, resulting in a light intensity of 14.6 μmol m$^{-2}$ s$^{-1}$ PPFD. Cells were

then imaged after 30 ($t_{30}$) and 90 ($t_{90}$) min of light exposure for phototactic behavior, with quantification based on the cell location relative to the light source. Freshly sampled swimming cells kept in the dark were used as a control. Furthermore, the same millimetric chambers were completely covered in aluminum foil, which was then pierced to leave a small circular area exposed to light. The chambers were incubated in the dark for 60 min, and then placed at 14 and 28 cm from the LED source (14.6 and 4.4 $\mu$mol m$^{-2}$ s$^{-1}$ PPFD, respectively) and cells imaged after 30 min.

To investigate the potential effects of the domestication process, the same experiment was also performed on laboratory *C. okenii* strain LaCa cells grown in the incubator. Cells were grown until their early exponential phase, to have the same physiological growth stage as the wild cells at the time of sampling [33]. Laboratory-grown cells in the same growth stage were kept in the dark and used as a control. In both experiments, distribution of cell in the different areas of the millifluidic chamber was determined by ImageJ automatic cell counting on the images obtained at the microscope. Light intensity was measured with a portable LI-180 spectrometer (LI-COR Biosciences, Lincoln, NE).

## Quantification of *C. okenii* adhesion

*C. okenii* cells maintained under anaerobic conditions were harvested using a 1 ml syringe equipped with a suitable needle. 0.5 ml of the cell solution was withdrawn from various sections of the cell suspension and subsequently centrifuged at 5,000 rpm for 60 s. Following centrifugation, 20 $\mu$l of the pellet was carefully transferred onto an agarose gel substrate and allowed to settle onto the substrate for a duration of 10 min. Subsequently, a tipless cantilever was calibrated and installed for liquid measurements. The cantilever was calibrated using force spectroscopy on a hard silica substrate in deionized water environment. The deflection sensitivity was calculated from the retraction slope of the force distance curve. The cantilever spring constant (0.2 N/m) was taken as provided and measured by NanoAndMore GmbH (Wetzlar, Germany). For the purpose of force-distance spectroscopic measurements, the cantilever was positioned in an area densely populated with cells. A grid of 20×20 measurement points, spaced 1 $\mu$m apart, was then recorded (at least 1,000 points were measured for each sample) for multiple replicates. A detailed description of the force-distance spectroscopic method employed here can be found in Ref. [New Ref.]

## Modeling mechanics and stability of swimming cells

We developed a cell-level swimming mechanics model to understand the role of the cell morphology and intracellular SGBs, and specifically, delineate the impact of these phenotypic alterations on the orientation stability of the swimming cells (the ability of cells to reorient back to the equilibrium swimming direction after they are perturbed). The model considers different forces and moments acting on a *C. okenii* cell (Table 1), by virtue of its propulsion, morphology and the SGBs number and intracellular distribution (S2B Fig in S1 File), establishing the factors which determine the cell's up-swimming stability (S3 Text in S1 File).

Bacteria cells have been traditionally modeled either as a spherocylinder or a spheroid geometry [34, 35]. We have thus simulated the drag for both the configurations and the difference between these values using COMSOL Multiphysics (the validation for the configuration of a sphere is presented in S3 Fig in S1 File). A maximum error of ~11% is observed between them. Since bacteria are strictly neither spherocylinders nor spheroids, the realistic error should be even less. For the sake of convenient representation without sacrificing the essential physics, we have considered the bacteria as a spheroid shape (S4 Text in S1 File).

**Table 1. List of parameters used for the computing swimming stability of *C. okenii* cells.**

| Parameters (Symbol) | Value | Unit | Ref. |
|---|---|---|---|
| Major radius (*a*, length of the cell) | 8 | μm | |
| Minor radius (*b*, width of the cell) | 2–5 | μm | |
| Velocity (*U*, swimming speed) | 16 | μm s$^{-1}$ | |
| Velocity angle (*θ*, direction of swimming) | $\pi/6$ | rad | |
| Medium viscosity (*η*) | $10^{-3}$ | Pa·s | |
| Specific gravity of cell ($\rho_{cell}$) | 1.01–1.10 | - | [36] |
| Density of sulfur globule ($\rho_O$) | 1.3 | g cm$^{-3}$ | [37] |
| Density of cytoplasm ($\rho_{cyt}$) | 1.05 | g cm$^{-3}$ | [38] |
| Density of medium ($\rho_{fluid}$) | 1.036 | g cm$^{-3}$ | |
| Sulphur globule radius ($r_O$) | 1.8 | μm | |

## Statistical analyses

Statistical analyses were performed with GraphPad Prism (version 9 for Windows, GraphPad Software, La Jolla, CA). Normality of the data was checked using the Shapiro-Wilk test. Two sample *t*-test was performed to compare laboratory-grown *C. okenii*'s cell volume and aspect ratio at the exponential growth stage with lake-sampled cells. The same statistical analysis was conducted to compare the number, size, and total volume accumulation of sulfur globules of natural and domesticated cells. Two-way ANOVA with Tukey's multiple comparisons correction test was used to compare the ratios of motile / non-motile cells in the phototaxis experiments.

## Results

### Domestication modifies cell morphology and intracellular SGBs attributes

To uncover the reason behind the differences observed in *C. okenii* motility between wild and domesticated populations, we looked for potential alterations of the morphological features of the cells. Cell phenotype and SGBs characteristics were monitored in the artificial growth conditions of the incubator (domesticated) and compared with cells freshly isolated from Lake Cadagno (Lake), using cell-level quantitative imaging. Laboratory-grown cells displayed an elongated shape, resulting in a considerably higher aspect ratio (length / width) in contrast to the more rounded shape (lower aspect ratio) maintained by natural samples (Table 2). Conversely, volume of lake-sampled cells was nearly 3-fold larger (207.6 ± 40.7 μm$^3$) than laboratory cells (69.9 ± 12.4 μm$^3$). Lake cells also displayed larger SGBs (radius length) than INC cells (Table 2). This reflected on the globules volume relative to the cell size ($V_{SGBs}/V_{cell}$, total

**Table 2. Mean values ± SD (in μm) of cell shape and volume, SGBs number, size, total volume per cell and relative density (in g cm$^{-3}$) for *C. okenii*.**

| | | INC | Lake |
|---|---|---|---|
| **Cell morphology** | Length | 9.36 ± 0.51 | 9.91 ± 1.73 |
| | Width | 3.35 ± 0.10 | 6.33 ± 0.38 |
| | Volume | 69.90 ± 12.47 | 207.66 ± 40.70 |
| **SGBs** | Number | 5 ± 1 | 6 ± 2 |
| | Radius | 0.29 ± 0.12 | 0.40 ± 0.08 |
| | $V_{SGBs}/V_{cell}$ | 0.013 ± 0.006 | 0.008 ± 0.003 |
| **Density** | + SGBs | 1.055 ± 0.004 | 1.052 ± 0.001 |
| | - SGBs | 1.000 ± 0.038 | 1.029 ± 0.007 |

globules volume/cell volume) shown in Table 1, which is significantly higher in the domesticated population ($p < 0.01$).

Variations in the specific content of SGBs exerted a major influence on cell density (Table 2). Particularly, effective cell density was significantly higher in the presence of intracellular SGBs than when SGBs were subtracted ($p < 0.01$), the difference being the density of the structural cell material. Such difference was more pronounced in laboratory cells (+ 5.5%), where the presence of SGBs increased cell density from 1.000 to 1.055 g cm$^{-3}$, coinciding with the larger SGBs size and $V_{SGBs}/V_{cell}$ (Table 2). Overall, the effective cellular mass density for the lab-grown cells was 0.3% higher than that of the cells from the lake (1.052 g cm$^{-3}$).

Alongside variations of the cellular morphology and SGBs attributes, observed that the domesticated *C. okenii* population exhibited fewer flagella compared to the lake population. Phase contrast micrographs track the presence of a polar flagella bundle on the cells freshly sampled from the lake chemocline (Fig 1A). In contrast, cells lacking flagella were detected in the domesticated samples (Fig 1B and S9 Fig in S1 File), even though the imaging conditions were the same for both samples.

## Absence of motility of *C. okenii* in laboratory environment

Variations in the swimming behavior resulting from the observed alterations in cell morphology were investigated by quantifying and comparing motility between domesticated and wild *C. okenii*. To do so, we defined three different motility regimes, 'no/low motility' ($< 5$ μm s$^{-1}$), 'medium motility' (5–20 μm s$^{-1}$), and 'high motility' ($> 20$ μm s$^{-1}$) and calculated the wild (Lake) and domesticated strain cell distribution among them (Fig 1C and S5 Fig in S1 File). Within one hour from the sampling, the wild population (Lake) showed higher motility than domesticated cultures, with 55% of the lake-sampled cells falling into the 'high motility' category compared to a value of 0% for the domesticated population in exponential phase (Fig 1C). Concomitantly to the reduction in motility, we recorded a change in adhesive interactions between lake and lab-grown *C. okenii* cells. We employed atomic force microscopy (AFM, see Materials and Methods) to measure such variation. Freshly isolated *C. okenii* cells (Lake) had a cell-surface adhesion of 0.211 ± 0.091 nN, whereas the cell-surface adhesion enhanced significantly for the laboratory incubated cells, by ~ 4-fold to 0.836 ± 0.584 nN (Fig 1D).

## Role of cell shape in swimming ability

Changes in cell morphology and density were related to swimming mechanics for both Lake and domesticated cell types. In cells sampled from the lake (Lake), SGBs tended to accumulate below the cell center of gravity ($C_H$; Fig 2A). The center of mass of the SGBs, $C_O$, was located below $C_H$. Since the position of $C_H$ coincides with the cell's center of buoyancy, $C_B$, which overlaps the center of gravity (Fig 2A), the accumulation of SGBs in the lower part of the cell made it slightly aft-heavy, the difference between the mass of SGBs in the fore and aft region of the cell being statistically significant (1.46 *vs* 1.26 × 10$^{-6}$ μg, $p < 0.05$). The low value of $L_W$ (distance from the $C_B$; S1 Table in S1 File) showed that SGBs were mainly scattered near the $C_B$, resulting in an average $L_W/a$ ratio of 0.039 (± 0.025), which places lake cells close to the boundary of the phase plot where the orientation stability switches (Fig 3A). The orientation stability provides a measure of the propensity of cells to orient back against the gravity direction when perturbed from the equilibrium swimming direction: higher the orientation stability, smaller is the time taken for a cell to reorient back to its vertical swimming direction.

Conversely, domesticated cells were characterized by a larger offset length of 0.41 (± 0.31) μm (S1 Table in S1 File), indicating a SGBs distribution farther from the cell $C_B$. Although it was not possible to distinguish the fore from the aft section of the cell for the domesticated

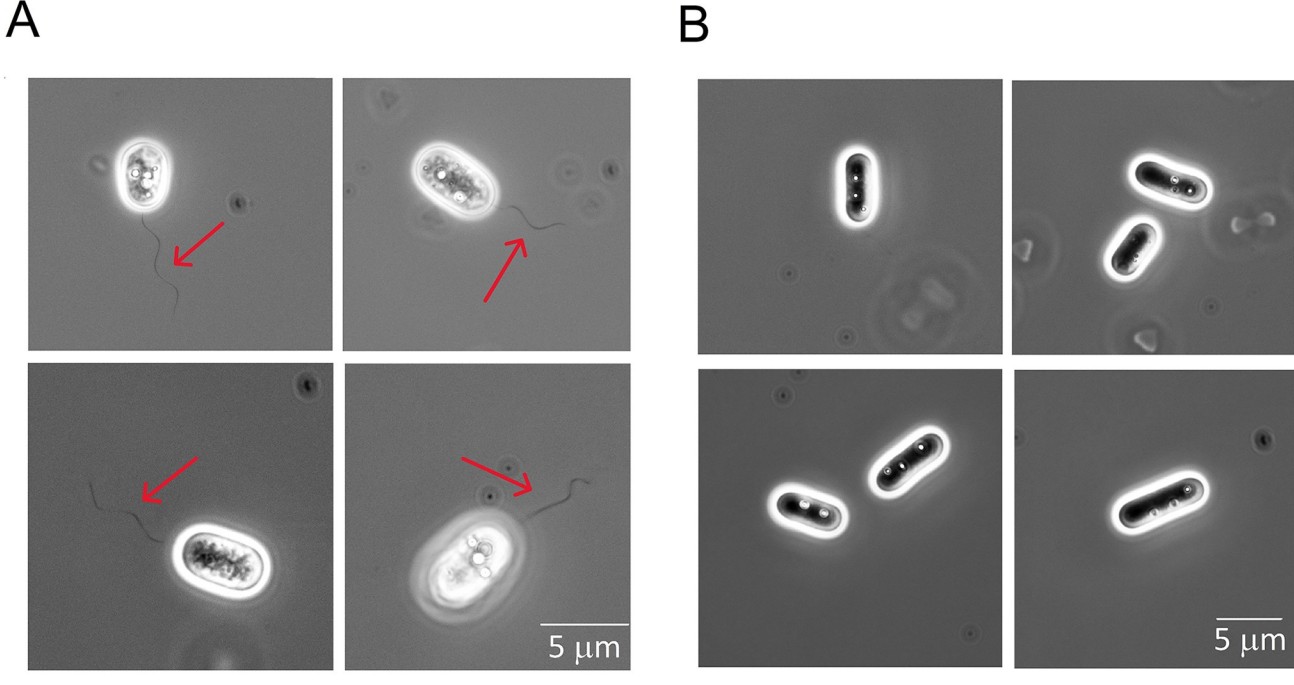

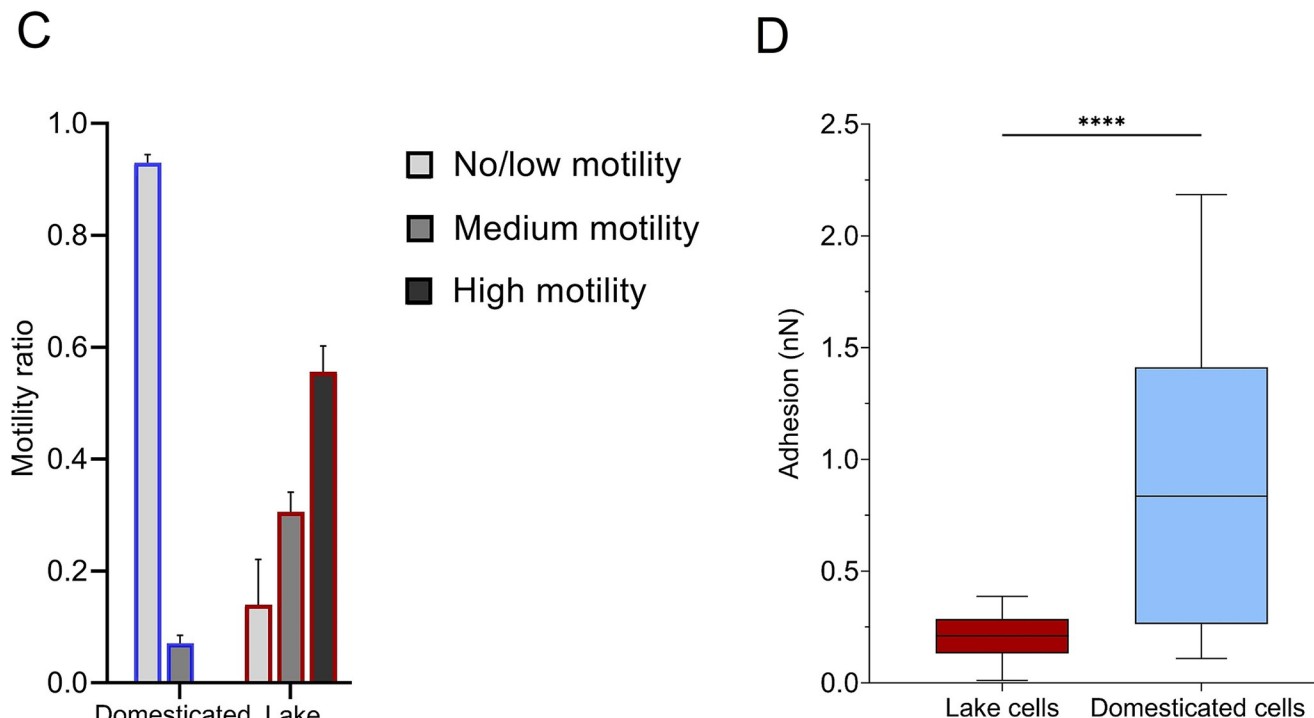

**Fig 1. Domestication alters cell morphology and SGB characteristics.** Microscopy images (100X, phase contrast) of *C. okenii* cells from (A) a fresh lake water sample and (B) domesticated laboratory-grown cells. No flagella are visible in domesticated cells. Intracellular sulfur globules are visible as highly refractive spheres. Red arrows indicate the polar flagellar tuft. (C) Bar plot of the different ratios of domesticated and natural cells within the three motility regimes. Blue and red outlines of the columns indicate domesticated and lake cells, respectively. Error bars represent standard deviation (N = 3). (D) *C. okenii* cells show enhanced adhesion after domestication. The boxplots illustrate the adhesion of cells to an agarose surface for lake cells freshly after

isolation (maroon-color); and the domesticated cells (light blue, stationary phase). The domesticated cells show a significantly high adhesion interaction with the surfaces. Unpaired *t* test, $p < 0.01$; asterisks indicate statistically significant difference.

populations, we did not observe significant differences in the globules number and mass distribution between the two halves of the cell ($p = 0.06$). Furthermore, both the SGBs relative to the cell size in exponential stage, and the larger offset length $L_W$, increased the rotational moment that biomechanically influences cell orientation [29].

In addition to the intracellular mass distribution of SGBs, the cell aspect ratio also played a role in shaping the swimming behavior of *C. okenii*. Domesticated cells had an overall higher aspect ratio compared to the lake phenotype (Table 2). This caused domesticated cells to experience greater drag when moving due to their elongated shape (Fig 2B and S4A Fig in S1 File), making motility even more energetically costly. In contrast, lake cells were characterized by a lower aspect ratio (Table 2) and exhibited a spherical geometry, which facilitated swimming as this morphology leads to a reduction in viscous drag (Fig 2B and S4A, S4B Fig in S1 File).

### Analysis of the phototactic behavior of *C. okenii*

The marked variations in the cell swimming behavior, and the concomitant increased cell-surface adhesion, reflected on the phototactic response (S6 Fig in S1 File), a key trait of *C. okenii* wild population. In general, we observed that light triggered a phototactic response in *C. okenii* wild cells (Lake) after a period of incubation in the dark (1 h), with a swimming speed that increased after exposure to the light source. On the contrary, domesticated cells showed almost no reaction to light exposure (Fig 4). Motile Lake-sampled cells kept in the dark (30 and 90 min) were used as a negative control.

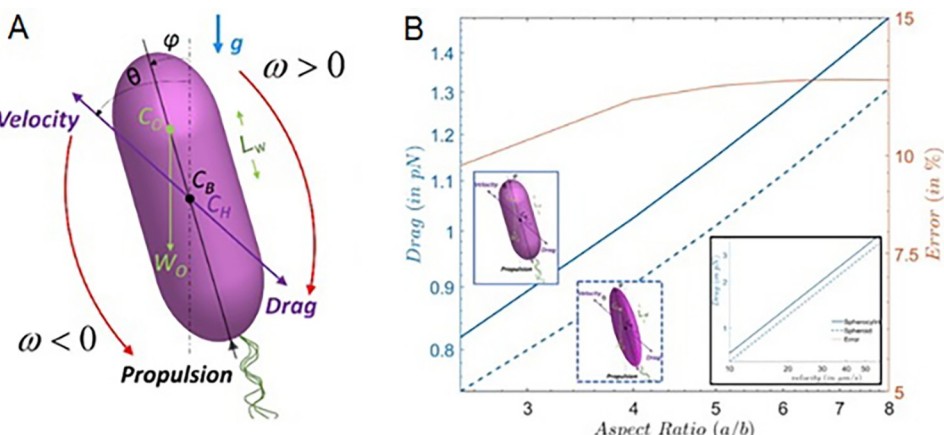

**Fig 2. Mechanics of *C. okenii* swimming.** (A) Schematics of the cell-level geometry for the formulation of the reduced-order model. The free-body diagram of all forces and torques (about point $C_B$) are color marked on the schematics. The swimming of the bacteria cell is considered to be stable when the cell rotates such that its pusher-type propulsion will propel the cell against gravity, $g$ (in the above configuration, this is achieved for $\omega > 0$). The weight and buoyancy forces act opposite to each other, to give an effective weight, $(\rho_{cell}-\rho_{fluid})Vg$, where $\rho_{cell}$ and $\rho_{fluid}$ respectively denote the cell and surrounding fluid densities, $V$ is the cell volume, $g$ is the acceleration due to gravity (acting downward, in the plane of the Fig). (B) Comparison of drag forces between spherocylinder and spheroid cell geometries for different cell aspect ratios. The y-axis on the right shows the error between the two estimations. For spheroid, the aspect ratio is the ratio between the minor axis and the major axis. For spherocylinders, the aspect ratio is the ratio between the radius of the spherical cap and half of the length of the central cylinder and radius of the spherical caps combined. The maximum error lies below ~11% for the two values. Alternative calculation of the spherocylinder aspect ratio can yield lesser error values (see S6 Text and S4 Fig in S1 File). Inset plot shows the drag force as a function of the swimming velocities.

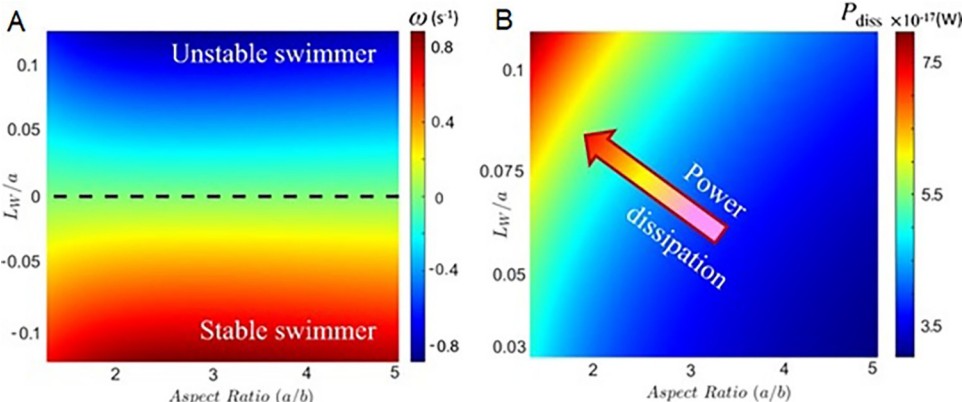

**Fig 3. Stability and energetics of *C. okenii* swimming.** (A) Phase plot presents the combined effect of cell aspect ratio ($a/b$) and the normalized offset length scale, $L_W/a$, ratio between the position of the cell center of weight (determined by the effective SGBs position) and its major axis. The dashed black line represents the boundary across which the orientation stability switches. Since, only the SGBs position dictates the swimming stability, the aspect ratio does not have any influence on the line of stability. However, the aspect ratio the drag on the cell, thereby determining the rotation rate, i.e., the time taken by the cell to attain equilibrium swimming direction. For a given $L_W/a$, a cell with higher aspect ratio will experience a higher rotation rate, and a higher degree of instability (stability) depending on whether it lies above (below) the dashed-black line, respectively. (B) Power dissipation by swimming *C. okenii* as a function of the aspect ratio and $L_W/a$ values obtained experimentally. For a given aspect ratio, the power dissipated increases with $L_W/a$, while for a given $L_W/a$, the dissipated power reduces with aspect ratio.

Wild cells (Lake) were exposed to light within a millifluidic confinement, half of which was covered to block incoming light. After 30 min, we observed a higher ratio of motile *vs* non-motile cells in the area exposed to light, compared to the dark half and the light-dark interface (Fig 4A). After 90 min, cells appeared to be less photo-responsive, with no significant differences in distribution across the millifluidic chip (Fig 4B) revealing an overall larger fraction of motile cells at $t_{30}$ than at $t_{90}$ (Fig 4). Similar phototactic behavior was observed in previously dark incubated cells after 30 min of localized LED illumination at two different light intensities (S7 Fig and S5 Text in S1 File). These observations are also supported by the notable differences in the way cells distributed throughout the millifluidic chamber (S8 Fig in S1 File). At $t_{30}$, *C. okenii* cells were significantly more abundant in the illuminated half of the chamber.

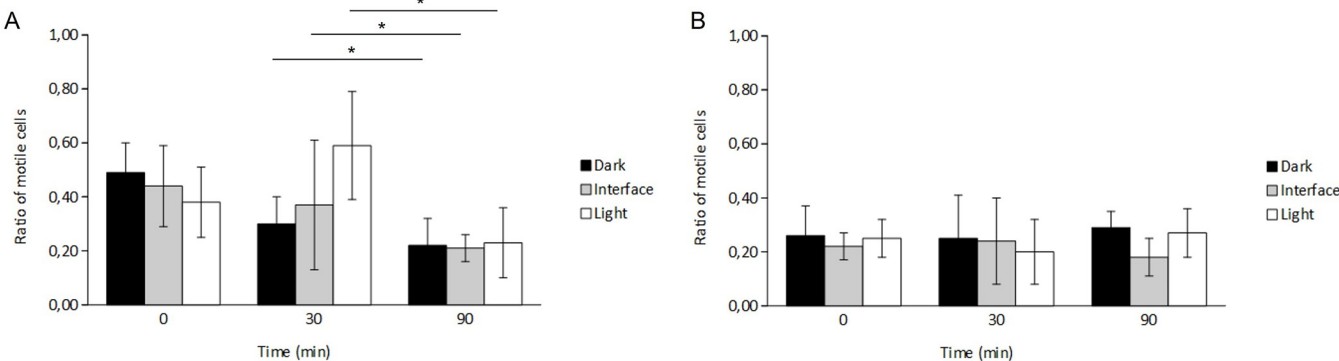

**Fig 4. Lake-sampled cells display phototactic behavior and higher motility than domesticated cells.** Ratios of motile *vs* non-motile cells for (A) lake-sampled and (B) laboratory-grown cells after 0, 30 and 90 min of light exposure in a half-shaded, half-illuminated microfluidic chip (see S6 Fig in S1 File). Two-way ANOVA, $p < 0.01$; post hoc Tukey test; asterisks indicate statistically significant differences. Error bars represent standard deviation (N = 3).

Conversely, no significant differences were found in the cell distribution at $t_{90}$ (S8A Fig in S1 File).

Instead, domesticated cells displayed almost no response to light when loaded in the same millifluidic chip. In fact, the absence of any significative difference in the motile *vs* non motile cell ratio, as well as in their distribution, across the three sections of the millifluidic device further confirms these observations (Fig 4B and S8B, S8C Fig in S1 File).

## Discussion

### Domestication drives synergistic phenotypic changes in *C. okenii*

Bacteria are highly responsive to environmental changes, leading to alterations in their morphology with important implications on cell motility [35, 39–41]. In our study, we observed significant differences in PSB *C. okenii*'s motility and morphology after adaptation to controlled laboratory conditions.

Domesticated populations exhibited smaller cell volumes compared to wild cells (Table 2), indicative of potential links between morphology and the demanding physical and energetic requirements associated with motility, with even a 0.2 μm change in cell diameter significantly escalating energy needs [42]. The relationship between morphology and motility becomes more evident when observing how aspect ratio influences the drag ($F_D$) a cell is subject to during motion [43]. In domesticated cells, due to their elongated shape (high aspect ratio), SGBs center of mass was farther from the cell geometric center (high $L_W$), and higher frictional coefficients due to increased surface area exceeded the reduction in cross-sectional area [44]. Conversely, wild *C. okenii* cells (Lake) maintained a rounded shape (low aspect ratio; Table 2) and SGBs were positioned around the cell geometric center (low $L_W$). Our COMSOL simulation findings were consistent with studies demonstrating higher $F_D$ for spherocylinders than spheroid-shaped objects [45, 46]. However, the direct relationship between SGBs size and cell shape remains uncertain, as specific SGBs localization in PSB species appears random, with few exceptions [47, 48]. Nonetheless, the content of intracellular sulfur inclusions significantly determined the buoyant density of *C. okenii* cells (Table 2). In fact, SGBs accumulation within the cell confinements increased the sedimentation component of swimming, making overcoming gravity more energetically expensive (S3 Text in S1 File). Elongated laboratory cells with SGBs during exponential growth were heavier than the rounded lake-sampled cells, explaining their reduced motility. This hypothesis is supported by the lower density of cells without globules (Table 2).

The implications extend to the natural environment, where *C. okenii* forms dense cell layers through coordinated swimming towards optimal light and sulfide conditions [49], leading to bioconvection [12]. While wild cells engage in such coordinated behavior, laboratory cells, nearly neutrally buoyant, sediment once SGBs form. Our cell density values align with those observed in *Chromatium* spp. by other studies [12, 50, 51]. Further insights emerge when considering that specific density is influenced by various regulated factors, including ribosomal material, proteins, and RNA, tailored to growth rate. However, the impact of growth rate on cell density is relatively modest (increase of about one unit in the second decimal) [52, 53], as volume gains likely offset greater cellular RNA and protein concentrations, contributing to the observed density variations in different growth environments. This nuanced interplay between cell aspect ratio, buoyancy and motility underscores the multifaceted impact of domestication on microbial behavior.

### Loss of flagellum and phototaxis in domesticated cells

Another key difference between wild (Lake) and domesticated *C. okenii* cells is the presence of a 20–30 μm long polar flagellar tuft (Fig 1A and S9 Fig in S1 File), consisting of about 40

flagella [54]. Given the energy-demanding nature of flagellar production, motile bacteria in culture media may lose motility [10, 22], prioritizing rapid growth over flagella assembly [9, 11]. Velicer *et al.* reported that in rod-shaped motile bacteria, like *Myxococcus xanthus*, motility can rapidly deteriorate under low selective pressure in artificial laboratory settings [55], while Barreto *et al.* observed that *Bacillus subtilis* can reduce swarming motility after domestication to laboratory conditions [4]. In contrast, natural environments with spatial organization and limited resource patches may enhance motility during evolutionary adaptation [56]. Considering our results, we hypothesize that domestication of *C. okenii* to laboratory conditions alters traits, such as cell geometry and the presence of flagella, essential for the fitness of wild populations. Overall, these changes in cell morphology suggest that, in laboratory cultures, motility is energetically costly and functionally redundant.

In addition to the capacity for active movement via flagellum, lab-domesticated cells also lose phototactic behavior (Fig 4). We compared the random motility of lake and domesticated cells, adapted to very different photoperiods and light intensities (S1A Fig in S1 File). The experiment revealed a gradual reduction in *C. okenii* swimming activity moving to conditions exhibiting considerable distinctions in terms of light regime. Simultaneously, in the laboratory cells, decreased motility (Fig 1C) was accompanied by increased phototrophic growth (Fig 1A). This suggests that, in controlled laboratory settings, cells may favor replication and growth over motility, an energetically costly process. Therefore, domestication can enhance fitness in the laboratory but may result in the loss of previous traits, such as motility, a phenomenon observed in various microbial strains like *Escherichia coli*, *Bacillus subtilis*, *Caulobacter crescentus*, and *Saccharomyces cerevisiae*[5–7].

In anaerobic phototrophic sulfur bacteria light is the principal factor driving motility and photosynthetic activity [57]. PSB utilize Bacteriochlorophyll *a* and *b*, whose absorbance spectra typically range from 750 to 800 nm in the near-infrared (NIR) region of the spectrum. In Lake Cadagno, very little to no NIR light penetrates in the water deep enough to reach the BL [32]. Light reaching this depth primarily consists of green wavelengths, falling within the absorption range of the predominant carotenoid, okenone (500–520 nm), typical of PSB [32]. This synergy between pigments enables PSB to efficiently utilize light energy, even in environments where NIR light is limited or absent.

Several studies reported that light, particularly in terms of photoperiod length, is a key parameter influencing cell activity in motile microorganisms, such as phytoplankton and bacteria [58–62]. In fact, when comparing laboratory cultures under the same light intensity and quality conditions in the incubator, reduced growth was observed in the 12/12 h photoperiod compared to the 16/8 h photoperiod (S10 Fig in S1 File).

Overall, these results again strongly suggest that growth, and subsequent adaptation, of *C. okenii* to laboratory conditions following propagation from the natural environment significantly alters cell motility.

## Adaptation to artificial settings impacts phototactic behavior

Wild *C. okenii* cells exhibited higher light-driven motility compared to domesticated ones (Fig 4A). Also, wild cells showed a higher number of motile cells after 30 min ($t_{30}$) of light exposure than after 90 min ($t_{90}$), (Fig 4A). This reduced motility at $t_{90}$ might result from an adaptive response to protect cells from photodamage [63, 64], as they were exposed to much higher light intensity (14.6 μmol m$^{-2}$ s$^{-1}$ PPFD) than in their natural habitat [32]. Similar behavior and motility reduction was observed when cells were exposed to a similar point light source (S7 Fig and S5 Text in S1 File). In contrast, domesticated *C. okenii* displayed a diminished response to light cues (Fig 4B). The observed reduced phototaxis of laboratory cells was

likely due to the artificial conditions in the growth environment [65], resulting in the adaptation to high light intensity, particularly in the cultivation chamber (38.9 µmol m$^{-2}$ s$^{-1}$ PPFD). Past research noted that motile PSB cells lost their phototactic ability when cultivated under continuous illumination and deposited as a thick red layer inside the culture vial [66, 67].

Wild *C. okenii* cells maintained consistent morphological traits, key factors determining the mechanics of swimming, across temperature variations (4˚C to 20˚C) from natural to laboratory environments. Conversely, domesticated cells, while retaining their morphology, exhibited reduced motility (S11 Fig in S1 File).

Overall, domesticated *C. okenii* strongly reduced their motility behavior due to the stable laboratory conditions, which differ from the fluctuating natural environment. The lack of fluctuations in essential growth parameters like light, nutrients, and temperature renders phototaxis non-essential for survival, contributing to the observed reduction of motility.

## Shifts in motility and enhancement of adhesion promote sessile lifeforms

The observed alterations of motility and phototactic behavior under laboratory conditions imply an adaptive response indicative of nuanced transition towards a sessile lifestyle. Cells experience the same dead torque rotation (i.e., the ability of a force to rotate an object about an axis perpendicular to the applied force), for same $L_W$/a and two different values of aspect ratio (a/b) (Fig 3A). This is because higher aspect ratios make the cell harder to rotate due to viscous resistance, but also subject it to higher torque. However, for a given aspect ratio, the dead torque rotation ω increases with an increase in the $L_W$/a ratio. Higher $L_W$/a or aspect ratio demands more active torque and, consequently, greater power dissipation, $P_{diss} = DU + R\omega\eta$, for up-swimming (Fig 3B). Thus, high $L_W$/a is associated with lower stability of the cells, indicating an increase in the time taken for the cells to reorient parallel to the gravity vector. Using COMSOL simulation, we compare spherocylinder and spheroid geometries (S4 Fig in S1 File). At low swimming velocities (low Reynolds numbers, Re), the $F_D$ differences between the two geometries are minimal. The error remains below 4% within the chosen velocity range, staying below 1% for the present system's velocity (up to ~20 µm s$^{-1}$). The validation includes a simulation of Stokes flow for a sphere, where drag force and Stokes drag coefficient ($C_D$) align well with analytical solutions (S3 Fig in S1 File).

This phenomenon suggests a reduced reliance on precise locomotion, emphasizing strategic adjustments that promote a shift from planktonic to sessile life forms under laboratory settings (Fig 1D). This transition is linked to the control of exopolysaccharide production and adhesin expression–critical factors in biofilm formation across diverse ecosystems [68]. The second messenger molecule cyclic dimeric guanosine monophosphate (c-di-GMP) plays a crucial role in this process [69–71]. For instance, in *Vibrio cholerae*, c-di-GMP governs cell shape dynamics by inhibiting the expression of the *crv*A gene, which is pivotal for curvature. This regulatory mechanism prompts a straight morphology, facilitating biofilm formation, whereas more rounded cells exhibit enhanced swimming capabilities [72]. Additionally, a separate study demonstrated that elongated *E. coli* cells exhibit better colonization of basal surface layers compared to spherical cells [5].

However, further research into *C. okenii*'s adhesive interactions, especially regarding potential biofilm formation and quorum sensing, is needed to shed light on the transition from motile to sessile states in these cells [New Ref.].

## Conclusion

Microorganisms adapt to changing environmental conditions through strategies including altered motility and morphology, spore formation, and biofilm production [73–76].

Particularly, motile species can migrate to better conditions but may lose motility in response to nutrient scarcity [77, 78] and extreme temperatures [79, 80], purposefully shedding flagella for survival [78]. Under the felicitous growth conditions of laboratory environments, traits like motility may become unnecessary, leading to flagella loss [81]. This adaptability, along with changes in the metabolic, physiological, and genetic profiles, precedes a transition from planktonic to sessile lifestyles in phototrophic bacteria, involving enhanced adhesion and the potential formation of biofilms [82, 83]. The formation of bacterial subpopulations following adaptation to new environments is frequently governed by epigenetic mechanisms, which create inheritable phenotypic diversity without modifying the DNA sequence [84]. These mechanisms exhibit considerable diversity, ranging from simple feedback loops to intricate, self-sustaining DNA methylation patterns [85, 86].

Such epigenetic mechanisms could potentially explain the marked differences in cell shape and swimming behavior of PSB *C. okenii* observed in our study, with domestication leading to a loss of motility and phototactic response. For instance, feedback loops might regulate the expression of genes involved in motility and adhesion, leading to the observed shift from a planktonic to a sessile state. Additionally, DNA methylation patterns could be responsible for the stable inheritance of these traits over multiple generations in the laboratory setting.

In their natural habitat, cells utilize sunlight and navigate the water column to optimize their position based on factors such as light, nutrients, and temperature. However, the laboratory environment presents challenges due to its inappropriate electromagnetic spectrum and higher temperatures. Our observations of domesticated PSB *C. okenii* suggest a synergistic shift in behavior and morphology towards a sessile lifestyle, possibly as a protective strategy dictated by epigenetic mechanisms to cope with the artificial conditions and ensure survival. This study focuses on phenotypic changes and motility suppression. Subsequent research will investigate the molecular and (epi)genetic mechanisms underlying these differences.

## Supporting information

**S1 File.**
(PDF)

## Acknowledgments

The authors thank Zornitza Tosheva from the Department of Physics and Materials Science of the University of Luxembourg and Pamlea Principi from the Institute of Microbiology (IM) of the University of Applied Sciences and Arts of Southern Switzerland (SUPSI) for the SEM pictures; Sofia Forner from the University of Insubria for the experiments using different photoperiods; Francesco Danza from the Section for Air, Water and Soil Protection (SPAAS) of Canton Ticino for fruitful discussion and constructive comments on the manuscript; the Alpine Biology Centre Foundation (CBA) for laboratory facilities and housing. The authors also acknowledge Robert Himelrick from University of Luxembourg for helping with the fabrication of experimental chambers and Max Fonseca from the Institute of Design (IDe) of the University of Applied Sciences and Arts of Southern Switzerland (SUPSI) for supporting with graphics.

## Author Contributions

**Conceptualization:** Francesco Di Nezio, Anupam Sengupta.

**Data curation:** Francesco Di Nezio, René Riedel, Arkajyoti Goshal, Jayabrata Dhar, Anupam Sengupta.

**Formal analysis:** Francesco Di Nezio, Irvine Lian Hao Ong, René Riedel, Arkajyoti Goshal, Jayabrata Dhar, Anupam Sengupta.

**Funding acquisition:** Anupam Sengupta.

**Investigation:** Francesco Di Nezio, Irvine Lian Hao Ong, René Riedel, Arkajyoti Goshal, Samuele Roman, Anupam Sengupta.

**Methodology:** Francesco Di Nezio, Anupam Sengupta.

**Project administration:** Anupam Sengupta.

**Resources:** Nicola Storelli, Anupam Sengupta.

**Software:** Arkajyoti Goshal, Jayabrata Dhar, Anupam Sengupta.

**Supervision:** Anupam Sengupta.

**Validation:** Nicola Storelli, Anupam Sengupta.

**Visualization:** Francesco Di Nezio, Nicola Storelli, Anupam Sengupta.

**Writing – original draft:** Francesco Di Nezio, Anupam Sengupta.

**Writing – review & editing:** Francesco Di Nezio, Irvine Lian Hao Ong, René Riedel, Arkajyoti Goshal, Jayabrata Dhar, Samuele Roman, Nicola Storelli, Anupam Sengupta.

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
