## [Decision Letter · Decision Letter 0]

25 Jun 2024

PONE-D-24-17650Phenotypic adaptations of motile purple sulphur bacteria Chromatium okenii during lake-to-laboratory domesticationPLOS ONE

Dear Dr. Sengupta,

Thank you for submitting your manuscript to PLOS ONE. After careful consideration, we feel that it has merit but does not fully meet PLOS ONE’s publication criteria as it currently stands. Therefore, we invite you to submit a revised version of the manuscript that addresses the points raised during the review process. In particular, both reviewers think that the differences between the two strains should be better characterised, including sequencing their genome.

We look forward to receiving your revised manuscript.

Kind regards,

Inês A. Cardoso Pereira, Ph.D.

Academic Editor

PLOS ONE

 [This work was supported by the Swiss National Science Foundation (grant number 315230–179264) and by the Institute of Microbiology (IM) of the University of Applied Sciences and Arts of Southern Switzerland (SUPSI) through the financing from the Department of “Socialità e Sanità” (DSS) of the Canton Ticino. I.L.H.O. thanks the support from Marie Skłodowska-Curie Actions Individual Fellowship (BIOMIMIC grant agreement number 897629). Support of the Luxembourg National Research Fund’s AFR-Grant (Grant no. 13563560), the ATTRACT Investigator Grant, A17/MS/11572821/MBRACE (to A.S.), and the FNR-CORE Grant (No. C19/MS/13719464/TOPOFLUME/Sengupta) are gratefully acknowledged.].  

5. In the online submission form, you indicated that [The data underlying the results presented in the study are available from the corresponding author upon reasonable requests.]. 

7.  We note that Figure S1, S2 and S9 in your submission contain copyrighted images. All PLOS content is published under the Creative Commons Attribution License (CC BY 4.0), which means that the manuscript, images, and Supporting Information files will be freely available online, and any third party is permitted to access, download, copy, distribute, and use these materials in any way, even commercially, with proper attribution. For more information, see our copyright guidelines: http://journals.plos.org/plosone/s/licenses-and-copyright.

a. You may seek permission from the original copyright holder of Figure S1, S2 and S9 to publish the content specifically under the CC BY 4.0 license. 

9. We notice that your supplementary figures are uploaded with the file type 'Figure'. Please amend the file type to 'Supporting Information'. Please ensure that each Supporting Information file has a legend listed in the manuscript after the references list.

Reviewers' comments:

Reviewer's Responses to Questions

**Comments to the Author**

1. Is the manuscript technically sound, and do the data support the conclusions?

Reviewer #1: Partly

Reviewer #2: Partly

2. Has the statistical analysis been performed appropriately and rigorously? 

Reviewer #1: Yes

Reviewer #2: I Don't Know

3. Have the authors made all data underlying the findings in their manuscript fully available?

Reviewer #1: Yes

Reviewer #2: Yes

4. Is the manuscript presented in an intelligible fashion and written in standard English?

Reviewer #1: Yes

Reviewer #2: Yes

5. Review Comments to the Author

Reviewer #1: The manuscript from Di Nezio et al describes phenotypic adaptation of Chromatium okenii, a motile phototrophic purple sulfur bacterium from meromictic Lake Cadagno, grown under laboratory conditions over multiple generations. They do this using different technologies arriving to the conclusion that naturally planktonic C. okenii switching to a sessile lifeform, together with changes in cell morphology, mass density, and distribution of intracellular sulfur globules. This is interesting, since in other species like B. subtilis and E. coli, domestication leads to loose capacity of biofilm formation.

I have several questions that the authors should consider.

General Comment:

The manuscript lacks clarity regarding the domestication process of Chromatium okenii. The terms "INC" and "domesticated" are used interchangeably (refer to Fig. 1C and D), which may confuse readers. Furthermore, it is mentioned in the Materials and Methods section (lines 116-117) that a strain has been cultured in the laboratory since 2016. It is critical to confirm whether this is the strain referred to as 'domesticated' throughout the study. For clarity and consistency, a direct comparison using the same strains pre- and post-domestication, similar to the approach taken in Bacillus subtilis research (Barreto 2020, ref 4), should have been done.

Specific Comments and Corrections:

Line 163: The magnification should be stated as 100x to match the legend in Figure 1.

Table 1: This table should be relocated to the results section, as it presents experimental data.

Line 223: Correct the typographical error from “onDetaito” to “on Detaito.”

Figure 1 (Panels A and B): The flagella are not visible, and the red arrows indicating them are difficult to discern. Similar visibility issues are noted in figure S9. Additionally, the labels for the upper and lower panels in figure S9 appear to be incorrect.

Figure 1 (Panel C): The blue and brown outlines around the column are not defined in the legend, which needs rectification for better understanding.

Line 308: The claim of an “increase in biofilm forming ability” needs verification. It is essential to distinguish between increased adherence and actual biofilm formation.

Line 341: The sentence should end after "reorient back to its vertical swimming direction;" the use of a comma here should be replaced with a period to correct the grammatical structure.

Reviewer #2: SUMMARY:

The authors compare cells of Chromatium okenii freshly removed from a lake (denoted ‘Lake’ or ‘wild’ cells) with cells of Chromatium okenii removed from the lake and kept in laboratory culture for several years (denoted ‘INC’ cells). The purpose reportedly is to investigate which phenotypic traits are altered in the laboratory culture. The observation that phenotypic adaptations take place in bacterial populations when transferred from nature to the laboratory – such as loss of motility – is not new. I am not convinced that the presented work provides sufficiently enough new knowledge to warrant publication. In order to justify publication, I think it is necessary to improve the comparison of the two populations, especially by identifying any genetic differences between the two populations, as well as comparing the two populations after they have been cultivated for a few generations under the exact same conditions in order to more clearly reveal the nature and dynamics of the adaptations in the lab strain.

GENERAL COMMENTS:

Introduction: we need a description of the ecophysiology of C. okenii so that the behaviors you are describing in the Results section can be understood and put into context. Especially: what are the observed physiological changes in the cells (e.g. cell size, sulfur globules size and abundance, motility, growth rate) as function of relevant environmental parameters (e.g. sulfide concentration, time of day, temperature). These are the cellular changes you are investigating in your experimental work, so it is important to state what the current knowledge is.

Lines 329-335: I do not see that the SGB accumulate above the cell center of gravity. The figure does not clearly show this. I think you need some quantitative measurements including statistics, possibly based on image analysis, if you want to make this claim.

Section on swimming behavior: I do not understand the importance of describing the mechanistic of swimming behavior of the wild cells, because the laboratory cells are not swimming at all. So exactly what the swimming mechanistic is if it is to be compared to no swimming at all, is not so important.

It is difficult to judge exactly what is causing the two populations under investigation to appear different because they have grown under very different conditions. The laboratory population has had more light, higher temperature, higher sulfide, higher nutrients, and no O2 exposure, when compared to the natural population. Thus, we cannot know if the observed differences are due to physiological adaptations that are reversible and how dynamic they are. A better way to investigate the nature of the adaptations may be to grow the two populations under the exact same conditions for a few generations and then perform your analyses.

If the authors want to contribute significantly to the understanding of the adaptations in the C. okenii population that has been removed from its natural environment and cultivated in under laboratory conditions for several years, I think the genetics of the two populations should be investigated. No such work is presented. As a minimum, I suggest: (1) A simple sequencing of the 16S rRNA gene and similar marker gene(s) to confirm whether the populations under investigation are closely related or not. (2) A genome sequencing of the two populations could confirm whether genetic changes have occurred. It may not be possible to identify all mutations that affect all relevant phenotypes for this work, but the level of genetic divergence could nevertheless be evaluated.

SPECIFIC COMMENTS:

Table 1: This table is in the Methods section and it has experimental results. It seems appropriate to put these data in the Results section.

Table 2: What are these parameters based on?

Line 293: “… we recorded a gradual loss of flagella…”. This statement indicates that you have observed the population numerous times over a long period and observed that the flagella are decreasing in number and/or size. I think this is not the case. You simply compare “Lake” cells and “INC” cells.

6. PLOS authors have the option to publish the peer review history of their article (what does this mean?). If published, this will include your full peer review and any attached files.

Reviewer #1: No

Reviewer #2: No

---

## [Author Response · Author response to Decision Letter 0]

27 Jul 2024

Reviewer #1

The manuscript from Di Nezio et al describes phenotypic adaptation of Chromatium okenii, a motile phototrophic purple sulfur bacterium from meromictic Lake Cadagno, grown under laboratory conditions over multiple generations. They do this using different technologies arriving to the conclusion that naturally planktonic C. okenii switching to a sessile lifeform, together with changes in cell morphology, mass density, and distribution of intracellular sulfur globules. This is interesting, since in other species like B. subtilis and E. coli, domestication leads to loose capacity of biofilm formation.

We are grateful to the Reviewer for the overall positive reception of our work. Below we have addressed the comments and questions raised. 

I have several questions that the authors should consider.

General Comment:

The manuscript lacks clarity regarding the domestication process of Chromatium okenii. The terms "INC" and "domesticated" are used interchangeably (refer to Fig. 1C and D), which may confuse readers. Furthermore, it is mentioned in the Materials and Methods section (lines 116-117) that a strain has been cultured in the laboratory since 2016. It is critical to confirm whether this is the strain referred to as 'domesticated' throughout the study. For clarity and consistency, a direct comparison using the same strains pre- and post-domestication, similar to the approach taken in Bacillus subtilis research (Barreto 2020, ref 4), should have been done.

We thank the Reviewer for the comment. We agree with this and have accordingly sorted out the confusion between the “INC” and “domesticated” terms. We also confirm that the strain cultured in the laboratory is the one referred to as ‘domesticated’ throughout the manuscript. 

The Chromatium okenii strain (LaCa) used in the experiment pre- and post-domestication is indeed the same, as it is the only strain of this species found in Lake Cadagno [see Luedin et al., 2019 (https://doi.org/10.1038/s41598-018-38202-1); Danza et al., 2018 (https://doi.org/10.1371/journal.pone.0209743); Decristophoris et al., 2009 (https://doi.org/10.4081/jlimnol.2009.16)].

Specific Comments and Corrections:

Line 163: The magnification should be stated as 100x to match the legend in Figure 1.

We thank the Reviewer for this suggestion. The magnification info amended as per reviewer’s advice (Line 160 in the revised manuscript file).

Table 1: This table should be relocated to the results section, as it presents experimental data.

We thank the Reviewer, we have moved Table 1 (now Table 2) to the Result section (Line 279 of the revised manuscript).

Line 223: Correct the typographical error from “onDetaito” to “on Detaito.”

The term the reviewer is referring to here is not a typographical error but the name of the company manufacturing the cantilever tip (Line 216).

Figure 1 (Panels A and B): The flagella are not visible, and the red arrows indicating them are difficult to discern. Similar visibility issues are noted in figure S9. Additionally, the labels for the upper and lower panels in figure S9 appear to be incorrect.

Figure 1 (Panel C): The blue and brown outlines around the column are not defined in the legend, which needs rectification for better understanding.

We thank the Reviewer for this observation. In Figure 1 and S9 we have now enhanced the contrast to better visualize the flagella. We also corrected the labels for the upper and lower panels in Figure S9. For Figure 1 (Panel C), we defined the column colored outlines in the caption.

Line 308: The claim of an “increase in biofilm forming ability” needs verification. It is essential to distinguish between increased adherence and actual biofilm formation.

To avoid confusion, we removed the aforementioned sentence.

Line 341: The sentence should end after "reorient back to its vertical swimming direction;" the use of a comma here should be replaced with a period to correct the grammatical structure.

We thank the Reviewer. The sentence was amended as suggested by the reviewer (Line 325).

Reviewer #2

SUMMARY: The authors compare cells of Chromatium okenii freshly removed from a lake (denoted ‘Lake’ or ‘wild’ cells) with cells of Chromatium okenii removed from the lake and kept in laboratory culture for several years (denoted ‘INC’ cells). The purpose reportedly is to investigate which phenotypic traits are altered in the laboratory culture. The observation that phenotypic adaptations take place in bacterial populations when transferred from nature to the laboratory – such as loss of motility – is not new. I am not convinced that the presented work provides sufficiently enough new knowledge to warrant publication. In order to justify publication, I think it is necessary to improve the comparison of the two populations, especially by identifying any genetic differences between the two populations, as well as comparing the two populations after they have been cultivated for a few generations under the exact same conditions in order to more clearly reveal the nature and dynamics of the adaptations in the lab strain.

We thank the Reviewer for their feedback and observations. Based on these, we have addressed the comments and questions raised. 

GENERAL COMMENTS:

Introduction: we need a description of the ecophysiology of C. okenii so that the behaviors you are describing in the Results section can be understood and put into context. Especially: what are the observed physiological changes in the cells (e.g. cell size, sulfur globules size and abundance, motility, growth rate) as function of relevant environmental parameters (e.g. sulfide concentration, time of day, temperature). These are the cellular changes you are investigating in your experimental work, so it is important to state what the current knowledge is.

Based on the reviewer’s valuable feedback, we have incorporated in the Introduction (see Lines 74-76, 82-84, 90-91) more details on the ecophysiology of Chromatium okenii to contextualize the behaviors described.

Lines 329-335: I do not see that the SGB accumulate above the cell center of gravity. The figure does not clearly show this. I think you need some quantitative measurements including statistics, possibly based on image analysis, if you want to make this claim.

We thank the Reviewer for this comment. We have amended the paragraph accordingly (lines 327-330 of the revised manuscript).

Section on swimming behavior: I do not understand the importance of describing the mechanistic of swimming behavior of the wild cells, because the laboratory cells are not swimming at all. So exactly what the swimming mechanistic is if it is to be compared to no swimming at all, is not so important.

We thank the Reviewer for this comment and regret that the point for the mechanistic model was not immediately clear. Using the cell-level analyses, we have attempted to present a cell mechanical model of swimming, specifically to show how: (i) internal distribution and dimensions of the sulphur globules and (ii) shape of the cells impact their ability to against the gravity direction (or toward light). Our analyses reveals that the change in the sulphur globule attributes, in combination with the change in the cell shape facilitates the cells’ swimming ability under lab conditions (with sufficient light and nutrient availability). 

It is difficult to judge exactly what is causing the two populations under investigation to appear different because they have grown under very different conditions. The laboratory population has had more light, higher temperature, higher sulfide, higher nutrients, and no O2 exposure, when compared to the natural population. Thus, we cannot know if the observed differences are due to physiological adaptations that are reversible and how dynamic they are. A better way to investigate the nature of the adaptations may be to grow the two populations under the exact same conditions for a few generations and then perform your analyses.

We are grateful to the Reviewer for bringing up this question. Our data indicate significant phenotypic differences between the natural populations of C. okenii from Lake Cadagno and those growing under laboratory conditions. These differences are characterized by several key observations on cell morphology and density, motility and surface attachment. Additionally, our computational model of cell mechanics confirms the role of these phenotypic shifts in suppressing the planktonic lifeform and promoting a sessile state.

These data suggest that the observed phenotypic changes are not merely a result of different environmental conditions but represent adaptive responses to the laboratory environment. This is supported by the consistency of these changes across multiple generations of laboratory culture (as suggested by the Reviewer in this comment, for instance please see Figures 1 and 4). 

The literature cited throughout the manuscript provides additional support for our observations. It is well-documented that microorganisms adapt to laboratory conditions through phenotypic changes, including loss of motility and enhanced surface attachment, driven by selective pressures in resource-replete environments [Cooper and Lenski, 2000 10.1038/35037572; Barrick and Lenski, 2013 10.1038/nrg3564]. Also, as reported in Lines 466-468, the high energy costs associated with maintaining motility appendages and their functions, such as flagella, can lead to their reduction or loss under laboratory conditions where selective pressures differ from natural environments [Velicer et al., 1998 https://doi.org/10.1073/PNAS.95.21.12376]. Furthermore, the observed phenotypic changes, including alterations in cell morphology and sulfur globule distribution, align with known adaptive strategies of bacteria transitioning from natural to artificial environments [Maughan et al., 2007 10.1128/AEM.00374-11].

If the authors want to contribute significantly to the understanding of the adaptations in the C. okenii population that has been removed from its natural environment and cultivated in under laboratory conditions for several years, I think the genetics of the two populations should be investigated. No such work is presented. As a minimum, I suggest: (1) A simple sequencing of the 16S rRNA gene and similar marker gene(s) to confirm whether the populations under investigation are closely related or not. (2) A genome sequencing of the two populations could confirm whether genetic changes have occurred. It may not be possible to identify all mutations that affect all relevant phenotypes for this work, but the level of genetic divergence could nevertheless be evaluated.

We appreciate the reviewer's insightful comments and agree that genetic analysis would provide valuable insights into the adaptations of Chromatium okenii populations cultivated under laboratory conditions compared to their natural counterparts. Our study has primarily focused on phenotypic adaptations and the mechanistic understanding of these changes. We have employed a range of techniques such as microfluidics, atomic force microscopy, quantitative imaging, and mathematical modeling to investigate these adaptations. 

We would like to highlight here that the genome of domesticated C. okenii LaCa (2022, six years after its initial isolation and sequencing) was recently re-sequenced; see Luedin et al., Scientific Reports 9, 1936, 2019. The results demonstrated no notable differences between the two genomes, which appear to be nearly identical. In light of this observation, we prioritized recording phenotypic changes over time, which we report here. We have now adapted the text and have proposed the role of epigenetic mechanisms (Lines 550-560 in the revised manuscript) as a possible reason for the phenotypic differences we have reported here. 

The inclusion of genetic analysis suggested by the Reviewer might not be adequate to establish molecular differences between the two populations. Comprehensive genetic analyses, including epigenetic changes may require significant resources and time, and thus are out of the current scope of this work. During the initial phase of our study, we prioritized phenotypic analysis to establish a foundation for understanding adaptive changes since the genome of the domesticated cell lines did not show any significant difference with their natural counterpart. The genomic and epigenetic analyses will be taken up in a forthcoming study.

SPECIFIC COMMENTS:

Table 1: This table is in the Methods section and it has experimental results. It seems appropriate to put these data in the Results section.

We thank the Reviewer for this suggestion. We moved Table 1 (now Table 2) to the result section (Line 279).

Table 2: What are these parameters based on?

The parameters related to the cells (i.e., length, width, velocity, velocity angle, sulfur globules radius) were derived from microscopic measurements and morphological studies of C. okenii. The viscosity of the medium is the value typical for water at room temperature. Specific gravity of the cell, density of sulfur globules, and density of cytoplasm were found in reference materials (now provided in the table). Density of medium was approximated to that of seawater.

Line 293: “… we recorded a gradual loss of flagella…”. This statement indicates that you have observed the population numerous times over a long period and observed that the flagella are decreasing in number and/or size. I think this is not the case. You simply compare “Lake” cells and “INC” cells.

We thank the Reviewer, based on this comment, we have modified the sentence specifying that we just compared the two populations (Lines 272-274).

---

## [Editor Report · Decision Letter 1]

6 Aug 2024

Phenotypic adaptations of motile purple sulphur bacteria Chromatium okenii during lake-to-laboratory domestication

PONE-D-24-17650R1

Dear Dr. Sengupta,

We’re pleased to inform you that your manuscript has been judged scientifically suitable for publication and will be formally accepted for publication once it meets all outstanding technical requirements.

Kind regards,

Inês A. Cardoso Pereira, Ph.D.

Academic Editor

PLOS ONE
---

## [Editor Report · Acceptance letter]

29 Aug 2024

PONE-D-24-17650R1 

PLOS ONE

Dear Dr. Sengupta, 

I'm pleased to inform you that your manuscript has been deemed suitable for publication in PLOS ONE. Congratulations! Your manuscript is now being handed over to our production team.

Kind regards, 

on behalf of

Dr. Inês A. Cardoso Pereira 

Academic Editor

PLOS ONE